# EM-Paste: EM-guided Cut-Paste for Image-level Weakly Supervised Instance Segmentation

## Abstract

We propose EM-PASTE: an Expectation Maximization (EM) guided Cut-Paste compositional dataset augmentation approach for weakly-supervised instance segmentation using only image-level supervision. The proposed method consists of three main components. The first component generates high-quality foreground object masks. To this end, an EM-like approach is proposed that iteratively refines an initial set of object mask proposals generated by a generic region proposal method. Next, in the second component, high-quality context-aware background images are generated using a text-to-image compositional synthesis method like DALL·E. Finally, the third component creates a large-scale pseudo-labeled instance segmentation training dataset by compositing the foreground object masks onto the original and generated background images. The proposed approach achieves state-of-the-art weakly-supervised instance segmentation results on both the PASCAL VOC 2012 and MS COCO datasets by using only image-level, weak label information. In particular, it outperforms the best baseline by +7.4 and +2.8 $mAP_{0.50}$ on PASCAL and COCO, respectively. Further, the method provides a new solution to the long-tail weakly-supervised instance segmentation problem (when many classes may only have few training samples), by selectively augmenting under-represented classes.

## 1 Introduction

The instance segmentation task aims to assign an instance label to every pixel in an image. It has been found in many applications on many real-world domains Hafiz & Bhat (2020), e.g., self-driving cars, AR/VR, robotics, etc. Standard approaches to solving this problem involve framing it as a per-pixel labeling problem in deep learning framework He et al. (2017); Hafiz & Bhat (2020).

Training of instance segmentation methods requires a vast amount of labeled data He et al. (2017). Getting a large labeled dataset with per-pixel instance labels is very expensive, requires significant human effort, and is also a time-consuming process. In order to tackle these issues, alternative approaches have been proposed. One direction involves utilizing synthetic data to train instance segmentation methods Richter et al. (2017); Hu et al. (2019); Ge et al. (2022a). However, they generally suffer from the sim2real domain gap, and expert knowledge is required to create synthetic environments Hodaň et al. (2019). A few other works have used object cut-and-paste Dwibedi et al. (2017); Ghiasi et al. (2021); Ge et al. (2022b) to augment training data for instance segmentation tasks. However, these methods require the availability of accurate foreground object masks, so that objects can accurately be cut before they are pasted. Acquiring these foreground masks may require extensive human efforts, which can make this line of work difficult to scale.

Weakly-supervised learning approaches have evolved as important alternatives to solving the problem. A few of these methods Khoreva et al. (2017); Liao et al. (2019); Sun et al. (2020); Arun et al. (2020); Hsu et al. (2019) involve using bounding boxes as a source of weak supervision. Bounding boxes contain important cues about object sizes and their instance labels. However, even bounding boxes are taxing to label. Another line of works Zhou et al. (2018); Zhu et al. (2019); Cholakkal et al. (2019); Ge et al. (2019); Hwang et al. (2021); Laradji et al. (2019); Kim et al. (2021); Ahn et al. (2019); Arun et al. (2020); Liu et al. (2020) explores using only image-level labels for learning instance segmentation. Due to the lack of segmentation annotations,

those works generally need to introduce object priors from region proposals Zhou et al. (2018); Zhu et al. (2019); Arun et al. (2020); Laradji et al. (2019). One approach involves utilizing signals from class activation maps Zhou et al. (2018); Zhu et al. (2019), yet those maps do not provide strong instance-level information but only semantic-level, and can be noisy and/or not very accurate. Another procedure involves generating pseudo-label from proposals and training a supervised model with pseudo-label as ground truth. Those methods can not generate high-quality pseudo-labels which hinders the supervised model performance.

In this work, we propose EM-Paste, a new weakly-supervised instance segmentation approach using only image-level labels. It consists of: First, "EM-guided Cut": we extract high-quality foreground object masks using an Expectation Maximization (EM)-like method to iteratively optimize the foreground mask distribution of each interested class and refine object mask proposals from generic region segmentation methods Maninis et al. (2016); Arbeláez et al. (2014); Qi et al. (2021). Then, we generate high-quality background images, by first captioning the source image, then passing them to text-to-image synthesis method (similar to Ge et al. (2022b)), e.g., DALL-E Ramesh et al. (2021); sbe; Ding et al. (2021) and stable diffusion Rombach et al. (2022). Finally, "Paste": we create a large labeled training dataset by pasting the foreground masks onto the original and generated context images (Figure 3).

We achieve state-of-the-art (SOTA) performance on weakly-supervised instance segmentation on the PASCAL VOC Everingham et al. (2010) and COCO dataset Lin et al. (2014a) using only image-level weak label. We outperform the best baselines by +7.3 and +2.8 mAP$_{0.50}$ on Pascal VOC and COCO datasets respectively. EM-Paste also provides a new solution to long-tail weakly-supervised instance segmentation problem on Pascal VOC dataset. Additionally, we also show that EM-Paste is generalizable to object detection task.

## 2 Related works

**Weakly Supervised Instance Segmentation** Since acquiring per-pixel segmentation annotations is time-consuming and expensive, many weakly supervised methods have been proposed to utilize cheaper labels. Existing weakly supervised instance segmentation methods can be largely grouped in two categories, characterized by labels that the algorithms can access during the training phase. The first line of works explores the use of bounding boxes as weak labels for instance segmentation tasks Khoreva et al. (2017); Liao et al. (2019); Sun et al. (2020); Arun et al. (2020); Hsu et al. (2019). Notably, Khoreva et al. (2017) generate pseudo-instance mask by `GrabCut+` and MCG Arbeláez et al. (2014), and Hsu et al. (2019) restraint the bounding box by tightness. Another series of works have also started using image-level labels as weak labels for instance segmentation tasks Zhou et al. (2018); Zhu et al. (2019); Cholakkal et al. (2019); Ge et al. (2019); Hwang et al. (2021); Laradji et al. (2019); Kim et al. (2021); Ahn et al. (2019); Arun et al. (2020); Liu et al. (2020). Notably, Zhou et al. (2018) utilizes class peak response, Ge et al. (2019) refines segmentation seed by a multi-task approach, Arun et al. (2020) improve the generated pseudo-labels by viewing them as conditional probabilities, and Kim et al. (2021) transfer semantic knowledge from semantic segmentation to obtain pseudo instance label. However, aggregating pseudo-label across multiple images remains largely unexplored.

**Data Augmentations for Instance Segmentation** In recent years, data augmentation has been an indispensable component in solving instance segmentation tasks Hafiz & Bhat (2020); He et al. (2017). Ghiasi et al. (2021) found that large-scale jittering plays an important role in learning a strong instance segmentation model, especially in a weakly-supervised setting. Dwibedi et al. (2017) proposed a new paradigm of data augmentation which augments the instances by rotation, scaling, and then pastes the augmented instances to images. Entitled Cut-Paste augmentation strategy can diversify training data to a very large scale. Empirical experiments Dwibedi et al. (2017); Ghiasi et al. (2021) have found that cut paste augmentation can lead to a major boost in instance segmentation datasets. These approaches require the presence of foreground object masks. So they can not be applied for weakly-supervised instance segmentation problems using only image-level labels. In contrast, our approach is designed to work with only image-level label information.

**Long-Tail Visual Recognition** Instance segmentation models usually fail to perform well in real-world scenarios due to the long-tail nature of object categories in natural images Gupta et al. (2019); Van Horn et al. (2018); He & Garcia (2009); Liu et al. (2019). A long-tail dataset consists of mostly objects from head

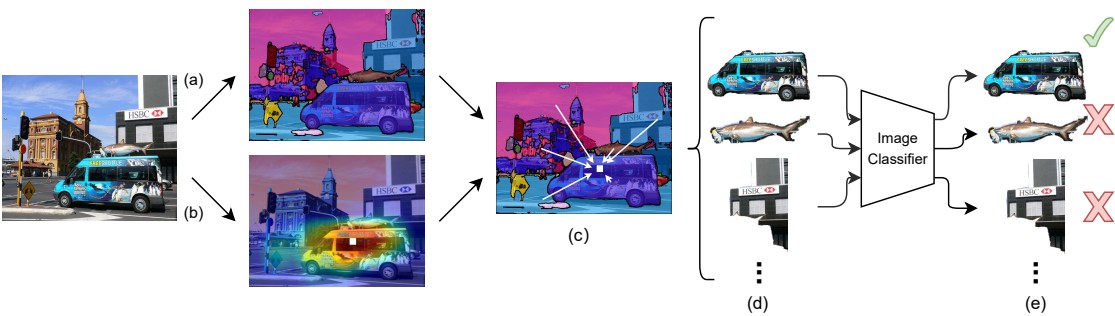

Figure 1: Step 1 of foreground extraction. (a) Entity Segmentation extracts segments from images. (b) Grad-CAM highlights a region based on the given label, and the center of moments (white dot on the image) is calculated for the highlighted region. (c) For all eligible segments, we compute the pixel-wise average distance to the center of the region highlighted by Grad-CAM. (d) We select $n$ segments that have the shortest distances to the center. (e) All $n$ foreground candidate segments are filtered using the classifier network, and we select the foreground with highest predicted probability.

classes, while objects from tail classes comprise relatively few instances. Existing instance segmentation methods He et al. (2017) often yield poor performance on tail classes, and sometimes predict head class all the time Wang et al. (2021). Existing methods to alleviate this include supervising models using a new loss that favors tail classes Wang et al. (2021); Hsieh et al. (2021) and dataset balancing techniques He & Garcia (2009); Mahajan et al. (2018) that re-distribute classes so that model can see more tail instances. However, few works evaluate weakly-supervised methods in long-tail setting.

## 3 Method

Our goal is to learn an instance segmentation model in a weakly supervised framework using only image-level labels. To this end, we propose EM-PASTE: EM-guided Cut-Paste with DALL·E augmentation approach that consists of three main components: foreground extraction (Section 3.1), background augmentation (Section 3.2), and compositional paste (Section 3.3). EM-PASTE produces an augmented dataset with pseudo-labels, and we train a supervised model using pseudo-labels as ground-truth.

### 3.1 EM-guided Foreground Extraction

We propose an Expectation Maximization (EM) guided foreground extraction (F-EM) algorithm. Given only image-level labels for a dataset, F-EM extracts as many high-quality foreground object masks as possible by iteratively optimizing the foreground mask distribution of each interested class and refining object mask proposals. There are three steps: 1) region proposal, 2) Maximization step to estimate object foreground distribution statistics of each interested object class, 3) Expectation step to refine the collection of matching region proposals given the approximated object foreground distribution statistics. Steps 2 and 3 are performed in an interactive manner. Figure 2 demonstrates different steps for extraction of foreground object masks.

**Step 1: Region Proposal.** In this step, the goal is to generate candidate foreground object segments corresponding to a given image label for each image. Suppose we are given a dataset $\mathcal{D} = \{(I_i, \mathbf{y}_i)\}_{i=1}^N$ where each image $I_i$ may contain one or more objects of different classes, therefore $\mathbf{y}_i$ is a binary vector that corresponds to image-level object labels for a multilabeled image $I_i$. We train an image classifier $f(\cdot)$ which takes image $I$ as input and predicts image label: $\mathbf{y} = f(I)$. Given a ground truth class $a$, for each image $I_i$ where $y_i^a = 1$, meaning that an object of class $a$ is present in image $I_i$, we generate the Grad-CAM Selvaraju et al. (2017) activation map through the classifier $f(\cdot)$ (Figure 1 (b)). Then we threshold the activation map to convert it to a binary mask $G_i^a$ that is associated with class $a$ and calculate x-y coordinate of the center of gravity of the object mask as: $c_i^a = (c_{ix}^a, c_{iy}^a) = (\sum_{x,y} G_i^a(x,y)x/G, \quad \sum_{x,y} G_i^a(x,y)y/G)$, where $G = \sum_{x,y} G_i^a(x,y)$. $c_i^a$ will be used as anchor to select foreground segments of class $a$ object for image $I_i$.

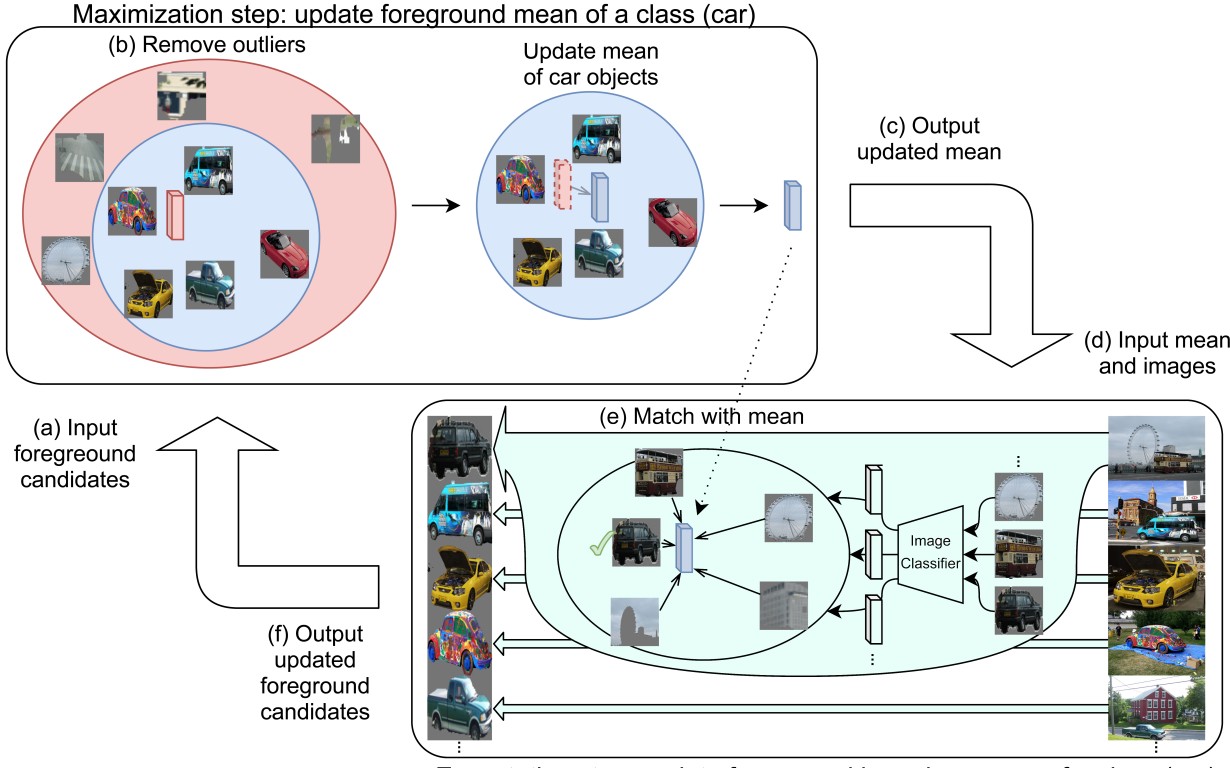

Figure 2: Step 2 and 3 of foreground extraction. (a) Each extracted foreground is passed to the classifier, and a latent representation of the image is extracted using a bottleneck layer. (b) Using the mean of all latent representations, we keep k% representations that are close to the mean and rule out outliers. (c) The mean is updated after ruling out the outliers. (d) For each image, latent representations of all eligible segments are obtained by the classifier network. (e) The segment with the highest cosine similarity to the updated mean is selected as the new foreground of the image. (f) After obtaining a new set of foregrounds, they are used as input of step 2 of the next iteration.

Next, for each input image $I_i$, we use an off-the-shelf generic region proposal method to propose candidate objects. The generic region proposal methods include super-pixel methods (SLIC Achanta et al. (2012), GCa10 Veksler et al. (2010); Felzenszwalb & Huttenlocher (2004)) and hierarchical entity segment methods (MCG Arbeláez et al. (2014), COBManinis et al. (2016), entity segmentation Qi et al. (2021)). These approaches only propose general class-agnostic segmentation masks with no class labels for the segments. In this work we show the results of using the entity segmentation method Qi et al. (2021) and COBManinis et al. (2016) to obtain a set of segments of the image $\mathcal{S}_i = \{s_i^1, s_i^2, ..., s_i^m\}$, but we note that our method is compatible to other methods as well.

Then we use the above computed Grad-CAM location anchor $c_i^a$ for interest class $a$ in image $I_i$ to find the correct foreground segment $O_i^a$ with label $a$ from $\mathcal{S}_i$. For each segment $s_i^j$, we calculate a pixel-wise average distance to the anchor $c_i^a$. We have the **location assumption** that the correct foreground segments should have a large overlap with the Grad-CAM mask $G_i^a$. In other words, foreground object mask $O_i^a$ for object class $a$ should have short average euclidean distance to the Grad-CAM center ($\text{dist}(O_i^a, c_i^a)$). We observe that this is generally true when there is only one object present in an image. But, in many images, more than one object from the same class can be present. In this case, foreground object may not perfectly overlap with the Grad-CAM activation map, because $c_i^a$ is the mean position of multiple foreground objects. To resolve this problem, we keep top-$n$ segments with the shortest distances to the center $c_i^a$. Typically $n \leq 3$. Then, we use the image classifier as an additional **semantic metric** to select the correct foreground. Specifically, we pass the top-$n$ segments through the same image classifier $f(\cdot)$ used in Grad-CAM. The segment with

---

**Algorithm 1** F-EM

---

    **Input:** Set of images $\mathcal{I} = \{I_i\}_{i=1}^N$, set of labels
$\mathcal{Y} = \{\mathbf{y_i}\}_{i=1}^N$, image classifier $f(\cdot)$ with feature extractor $\phi(\cdot)$, class of interest $a$

    **Output:** Set of foregrounds $\mathcal{O} = \{O_i\}_{i=1}^N$ of class $a$ objects.

  1:  $\mathcal{O} \leftarrow \emptyset$                                                                        ▷ Step 1

  2:  **for all** image $I_i \in \mathcal{I}$ where $y_i^a = 1$ **do**

  3:      $\mathcal{S}_i \leftarrow \text{EntitySeg}(I_i)$

  4:      $c_i^a \leftarrow$ center of Grad-CAM$(I_i, a)$

  5:      $p_s \leftarrow f(s)$ for $n$ of $s \in \mathcal{S}_i$ with smallest $\text{dist}(s, c_i^a)$

  6:      $\mathcal{O} \leftarrow \mathcal{O} \cup \{\arg\max_s\{p_s\}\}$

  7:  **for** $j$ iterations **do**                                               ▷ Step 2 (Maximization)

  8:      $\hat{\mu}, \hat{\Sigma} \leftarrow$ mean and covar of $\{\phi(O_i)\}_{i=1}^N$

  9:      $\mathcal{O}' \leftarrow$ k% of $O_i \in \mathcal{O}$ with smallest m-dist$(\phi(O_i), (\hat{\mu}, \hat{\Sigma}))$

10:      $\hat{\mu}', \hat{\Sigma}' \leftarrow$ mean and covar of $\{\phi(O_i')\}_{i=1}^{N \times k\%}$

11:      $\mathcal{O} \leftarrow \emptyset$                                                           ▷ Step 3 (Expectation)

12:      **for all** image $I_i \in \mathcal{I}$ where $y_i^a = 1$ **do**

13:          $\mathcal{S}_i \leftarrow \text{EntitySeg}(I_i)$

14:          $\mathcal{O} \leftarrow \mathcal{O} \cup \{\arg\min_{s \in \mathcal{S}_i} \text{m-dist}(\phi(s), (\hat{\mu}', \hat{\Sigma}'))\}$

---

the highest predicted probability is our initial selection of the foreground of the input image $O_i$. While, the initial extraction is far from perfect, because grad-cam localize the most discriminative location depends only on high-level classification information, which may have mismatch to the correct object location given complex scene.

To resolve the above issues and to further improve foreground object masks, we propose an iterative approach whereby we select a subset of segments $\mathcal{O}$ from the larger set of original segments $\mathcal{S}$ generated by the region proposal method. These selected segments are considered as foreground object segments. We frame the iterative segment selection within an Expectation-Maximization like steps.

The following EM steps assume that the latent representation (through a feature extractor $\phi(\cdot)$ of the image classifier $f(\cdot)$) of all foreground objects from the same class $a$ follow a distribution $p_{\psi_a}$, here we assume $p_{\psi_a} \sim \mathcal{N}(\mu, \Sigma)$ follows Multivariate Gaussian distribution because in the latent space of image classifier $f(\cdot)$, representations of images from same class should be a single cluster. Our goal is to find the optimal parameters of the distribution. This corresponds to generating right object masks. We follow an expectation maximization (EM-) like approach to find optimal parameters. This involves iteratively optimizing the distribution parameter $\psi_a$ which includes $\mu \in \mathbb{R}^d$ and $\Sigma \in \mathbb{R}^{d \times d}$ for each interest class $a$ (Maximization step). Then use $\psi_a$ to find accurate foreground segments in the latent space of $\phi(\cdot)$ (Expectation step). Figure 2 shows the whole process.

**Step 2 (M-step) Maximization.** In M-step, for each interest class $a$, our goal is to find the optimal parameters $\mu_a$, $\Sigma_a$ given the candidate foreground proposals $\mathcal{O}$. Here $\mathcal{O}$ are extracted foreground objects (from step 1, or step 3 at the previous iteration) for a specific class a. For each segment $O_i$, we generate its latent space representation $h_i$ by passing it through the image classifier $h_i = \phi(O_i)$. In particular, $h_i$ is the feature after the last convolution layer of the classifier. We compute $\hat{\mu} = \mathbb{E}(\psi) = \frac{1}{N}\sum_{i=1}^N h_i$ and $\hat{\Sigma} = \mathbb{E}((h - \hat{\mu})(h - \hat{\mu})^T)$ as initial mean vector and covariance matrix of latent space representations of all selected foreground segments. Because not all foreground masks $O_i$ are correct foregrounds, some of them may be background objects or objects with a different label. To remove the outlier and update the mean vector, we rule out outliers by keeping only $k\%$ of the segments that are closest to $\hat{\mu}$ based on the Mahalanobis distance m-dist$(h_i, (\hat{\mu}, \hat{\Sigma})) = \sqrt{(h_i - \hat{\mu})^T \hat{\Sigma}^{-1} (h_i - \hat{\mu})}$ of their latent representations. Using only the remaining foreground (inlier) segments, we compute a new mean $\hat{\mu}'$ and covariance matrix $\hat{\Sigma}'$ of the foreground object latent representations of the given class, which can be used to match more accurate foreground mask in E-step.

**Step 3 (E-step) Expectation.** In E-step, we regenerate the set of foreground segments $\mathcal{O}$ of class a by matching segment candidates with the updated $\hat{\mu}'$ and $\hat{\Sigma}'$ in M-step. In other words, we compute the "expectation" of the foreground mask for each image: $\mathbb{E}(O_i|\hat{\mu}', \hat{\Sigma}', I_i)$. For each image $I_i$, we start with the set of all eligible segments $\mathcal{S}_i$ (computed in step 1) again and generate the corresponding latent space representations $\Phi_i = \{h_i^1, h_i^2, ..., h_i^m\}$ as described earlier. We then compute a Mahalanobis distance (m-dist) between each latent representation of the segment $h_i^j$ and the new mean $\hat{\mu}'$ obtained from step 2. The segment with the smallest m-dist is selected as the new foreground of the image. With a new set of foreground segments $\mathcal{O}$, we can perform step 2 followed by step 3 again for, typically 2 or 3, iterations.

Algorithm 1 shows the details of the EM-guided Foreground Extraction algorithm.

## 3.2 Background (Context) Augmentation

Next step involves generating a large set of high-quality context images that could be used as background images for pasting foreground masks. One possible approach would be to use randomly selected web images as background images. However, prior works Dvornik et al. (2018); Divvala et al. (2009); Yun et al. (2021) have shown that context affects model's capacity for object recognition. Thus, selecting appropriate context images is important for learning good object representation, and thus beneficial for instance segmentation as well. To this end, we use a similar pipeline as DALL-E for Detection Ge et al. (2022b) to use image captioning followed by text-to-image generation methods to automatically generate background images that could provide good contextual information. More details in appendix.

**Image Captioning** Given an training set image, we leverage an off-the-shelf self-critique sequence training image captioning method Rennie et al. (2017) to describe the image, but we note that our method is agnostic to any specific image captioning method. These descriptions can capture the important context information. We further design a simple rule to substitute the object words, that has overlap with target interest class (in VOC or COCO) with other object words, in captions, to decrease the possibility of generating images that contains interest object, since they come without labels.

**Image Synthesis** We use the captions as inputs to text-to-image synthesis pipeline DALL · E Ramesh et al. (2021)[1], to synthesize a large set of high-quality images that capture all relevant contextual information for performing paste operation (Section 3.3). For each caption, we generate five synthesized images. Note that with our caption pruning rule described above, we assume that synthesized images do not contain foreground objects.

## 3.3 Compositional Paste

After foreground extraction (Section 3.1), we have a pool of extracted foregrounds where each class has a set of corresponding foreground objects. After background augmentation (Section 3.2) we have both the original background images and contextual augmented background images by DALL · E. We can create a synthetic dataset with pseudo instance segmentation labels by pasting the foreground masks onto the background images. For each background image, we select $n_p$ foregrounds based on a pre-defined distribution $p$, discussed later, and the goal is to paste those extracted foregrounds with the appropriate size. The appropriate choice of $n_p$ depends on the dataset. To force the model to learn a more robust understanding, each pasted foreground undergoes a random 2D rotation and a scaling augmentation. In addition, we note that direct object pasting might lead to unwanted artifacts, also shown in the findings of Dwibedi et al. (2017) and Ghiasi et al. (2021). To mitigate this issue, we apply a variety of blendings on each pasted image, including Poisson blurring, Gaussian blurring, no blending at all, or any combination of those. In practice, we find Gaussian blurring alone can yield sufficiently strong performance. Now we present two methods, each with their edges, of how to find the paste location. We leave the end-user to decide which method to use.

**Random Paste** In this simple method, we iteratively scale the foreground object by a random factor $\sim \text{Uniform}(0.3, 1.0)$, and paste in a random location on the image. We find a factor $> 1$ generally creates objects too large and a small factor enhances model learning capacity for small objects.

**Space Maximize Paste** This dynamic pasting algorithm tries to iteratively utilize the remaining available background regions to paste foreground objects. Our intuition is to force the pasted foregrounds to occupy

---

[1]In implementation, we use Ru-DALL · E sbe.

(a) The original image to be pasted.

(b) The **red circle** is the max inscribing circle found based on contour, denoted by the **blue line**.

(c) The first object, a person, is pasted on this image.

(d) After repeatedly applying above steps, four objects are pasted on the image.

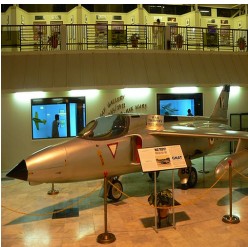 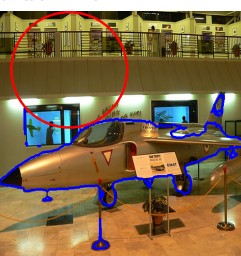 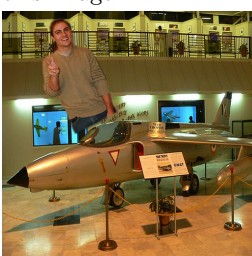 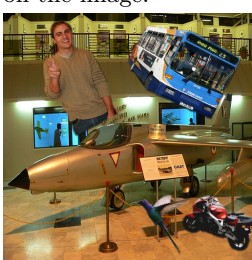

Figure 3: Illustrative example of Space Maximize Paste algorithm. In this example, four foreground objects are pasted on the background image that contains an aeroplane. In part (b) the max inscribing circle is found from contour based on region without aeroplane. We emphasize that the contour is found only based on image level, using process described in Section 3.1. Note that the person is scaled to match the size of the circle found in part (b), and a random rotation is performed.

as many spaces of the pasted background as possible, while remaining non-overlapping with the new to-be-pasted foreground and original background plus already pasted foregrounds. We give an illustrative example in Figure 3. Firstly, we find background regions where no object lies by computing the maximum inscribing circle from the contour of background images without existing foregrounds, original or pasted, as shown in the **red circle** in Figure 3b. The maximum inscribing circle gives a maximum region not occupied by any objects, thus providing the largest empty space. Next, we scale the pasted foreground to largely match the size of the radius of the maximum inscribing circle, rotate by a random degree, and paste to the location of the center of the maximum inscribing circle, shown in Figure 3c. We iteratively repeat the above steps to paste all $n_p$ foregrounds (Figure 3d). We note that since this method finds the background space with the decreasing area, thus able to synthesize images pasted with objects of various sizes.

**Selection Probability** The pre-defined selection distribution $p$ is crucial in that it imposes the class distribution of synthetic dataset produced by the paste method. We investigate and provide two types of probability to end users. The simplest type is a uniform distribution, i.e., selecting each image from foreground pool with the same chance. With this choice, the synthetic data approximately follows the class distribution of foreground pool. The second type is a balanced sampling, i.e. giving the classes with more instances a smaller weight to be selected while giving the classes with less instances a larger weight. This type enforces each class to appear in synthetic data in approximately the same quantity. In Section 4.4 we show that this setting is beneficial for long-tail problem.

# 4 Experiments

We demonstrate the effectiveness of EM-PASTE in weakly-supervised instance segmentation from image-level labels on Pascal VOC and MS COCO datasets. Additionally, we also show that EM-PASTE is generalizable to object detection task and highlight benefits of EM-PASTE in handling long-tail class distribution with only image-level label information.

## 4.1 Experiment Setup

**Dataset and Metrics** We evaluate EM-PASTE on Pascal VOC (Everingham et al., 2010) and MS COCO (Lin et al., 2014b) datasets. Pascal VOC consists of 20 foreground classes. Further, following common practice of prior works (Ahn et al., 2019; Arun et al., 2020; Sun et al., 2020), we use the augmented version (Hariharan et al., 2011) with 10,582 training images, and 1,449 val images for Pascal VOC dataset. MS COCO dataset consists of 80 foreground classes with 118,287 training and 5,000 test images. Per the standard instance segmentation and object detection evaluation protocol, we report mean average precision (mAP) (Hariharan et al., 2014) on two different intersection-over-union (IoU) thresholds, namely, 0.5 and 0.75. We denote these two mAPs as $\text{mAP}_{0.50}$ and $\text{mAP}_{0.75}$, respectively.

Table 1: Metrics for instance segmentation models on Pascal VOC 2012 val set. Here $\mathcal{F}$ means fully supervised, $\mathcal{B}$ and $\mathcal{I}$ mean bounding box and image level label based weakly supervised methods respectively. We highlight the best mAP with image level label in green , and bounding box label in blue . Our method outperforms prior SOTA image level methods. Further our method achieves better performance than some of the prior bounding box SOTA, although bounding box method has access to a lot more information about object instances.

| Method | Supervision | Backbone | mAP$_{0.50}$ | mAP$_{0.75}$ |
|---|---|---|---|---|
| Mask R-CNN (He et al., 2017) | $\mathcal{F}$ | R-101 | 67.9 | 44.9 |
| SDI (Khoreva et al., 2017) | $\mathcal{B}$ | R-101 | 44.8 | 46.7 |
| Liao et al.Liao et al. (2019) | $\mathcal{B}$ | R-50 | 51.3 | 22.4 |
| Sun et al.Sun et al. (2020) | $\mathcal{B}$ | R-50 | 56.9 | 21.4 |
| ACI Arun et al. (2020) | $\mathcal{B}$ | R-101 | 58.2 | 32.1 |
| BBTP (Hsu et al., 2019) | $\mathcal{B}$ | R-101 | 58.9 | 21.6 |
| PRM (Zhou et al., 2018) | $\mathcal{I}$ | R-50 | 26.8 | 9.0 |
| IAM (Zhu et al., 2019) | $\mathcal{I}$ | R-50 | 28.8 | 11.9 |
| OCIS (Cholakkal et al., 2019) | $\mathcal{I}$ | R-50 | 30.2 | 14.4 |
| Label-PEnet (Ge et al., 2019) | $\mathcal{I}$ | R-50 | 30.2 | 12.9 |
| CL (Hwang et al., 2021) | $\mathcal{I}$ | R-50 | 38.1 | 12.3 |
| WISE (Laradji et al., 2019) | $\mathcal{I}$ | R-50 | 41.7 | 23.7 |
| BESTIE (Kim et al., 2021) | $\mathcal{I}$ | R-50 | 41.8 | 24.2 |
| JTSM (Shen et al., 2021a) | $\mathcal{I}$ | R-18 | 44.2 | 12.0 |
| IRN (Ahn et al., 2019) | $\mathcal{I}$ | R-50 | 46.7 | 23.5 |
| LLID (Liu et al., 2020) | $\mathcal{I}$ | R-50 | 48.4 | 24.9 |
| PDSL (Shen et al., 2021b) | $\mathcal{I}$ | R-101 | 49.7 | 13.1 |
| ACI (Arun et al., 2020) | $\mathcal{I}$ | R-50 | 50.9 | 28.5 |
| BESTIE + Refinement (Kim et al., 2021) | $\mathcal{I}$ | R-50 | 51.0 | 26.6 |
| EM-Paste (Ours) | $\mathcal{I}$ | R-50 | 56.2 | 35.5 |
| EM-Paste (Ours) | $\mathcal{I}$ | R-101 | 58.4 | 37.2 |

**Synthesized Training Dataset** We do not touch on the segmentation label but instead generate pseudo-labeled synthesized training dataset using methods described in Section 3. Pascal VOC training dataset of 10,582 images consists of 29,723 objects in total, we extract 10,113 masks of foreground segments (34.0%). Similarly, MS COCO training set of 118,287 images consists of 860,001 objects in total, we extract 192,731 masks of foreground segments (22.4%) [2]. We observe that such masks are not perfect and contain noise, but overall have sufficient quality. To further ensure the quality of foregrounds, we filter the final results using 0.1 classifier score threshold. Additionally, we leverage image captioning and DALL·E (Section 3.2) to further contextually augment backgrounds. We generate 2 captions per image[3], synthesize 10 contextual backgrounds per caption, and utilize CLIP (Radford et al., 2021) to select top 5 backgrounds among the 10 synthesized images, together producing 10k and 118k augmented backgrounds for VOC and COCO respectively. To make the best use of both original backgrounds and contextually augmented backgrounds, we blend them together as our background pool, on which we apply methods from Section 3.3 to paste these extracted foregrounds. For simplicity, we use Random Paste method. We duplicate the original backgrounds twice to make the distribution between real and synthetic backgrounds more balanced.

**Model Architecture and Training Details** We train Mask R-CNN (He et al., 2017) with Resnet 50 (R-50) or Resnet 101 (R-101) as backbone (He et al., 2016). We initialize Resnet from ImageNet (Deng et al., 2009) pretrained weights released by `detectron2` (Wu et al., 2019). We deploy large-scale jittering (Ghiasi et al., 2021), and additionally augment training data with random brightness and contrast with probability 0.5. We run our experiments on one 32GB Tesla V100 GPU with learning rate 0.1 and batch size 128.

### 4.2 Weakly-supervised Instance Segmentation

In our setting, we follow the details in Section 4.1 and assume access to only image-level labels. That is, we *do not* use any segmentation annotation from the training set. We report VOC performance in Table 1.

---

[2]The number of per-class foreground masks extracted in Suppl.
[3]We augment each of 10,582 VOC image, and a random 10% sample of 118k COCO images.

Table 2: Weakly supervised instance segmentation on COCO val2017. Models here use image-level label.

| Method | Backbone | $mAP_{0.50}$ | $mAP_{0.75}$ |
|---|---|---|---|
| WS-JDS (Shen et al., 2019) | VGG16 | 11.7 | 5.5 |
| JTSM (Shen et al., 2021a) | R-18 | 12.1 | 5.0 |
| PDSL (Shen et al., 2021b) | R-18 | 13.1 | 5.0 |
| IISI (Fan et al., 2018) | R-101 | 25.5 | 13.5 |
| LLID (Liu et al., 2020) | R-50 | 27.1 | 16.5 |
| BESTIE (Kim et al., 2021) | R-50 | 28.0 | 13.2 |
| EM-PASTE (Ours) | R-50 | **30.8** | **20.7** |

**Baselines** We compare against previous weakly-supervised SOTA. Notably, Zhou et al. (2018) utilize peak response maps from image-level multi-label object classification model to infer instance mask from pre-computed proposal gallery; Laradji et al. (2019) generate pseudo-label training set from MCG (Arbeláez et al., 2014) and train a supervised Mask-RCNN (He et al., 2017); and Khoreva et al. (2017) generate pseudo-label using `GradCut+` and MCG (Arbeláez et al., 2014).

**Results** Quantitative results on Pascal VOC dataset have been shown in Table 1. Firstly, we experiment with the choice of R-50 or R-101. With more capacity brought by a deeper model, we find that R-101 works better compared to R-50, leading to +2.2 $mAP_{0.50}$ and +1.7 $mAP_{0.75}$ improvement. This validates that EM-PASTE is suitable for instance segmentation task. Secondly, our method can significantly outperform previous image-level SOTA by +7.4 $mAP_{0.50}$ (from 51.0 to 58.4), and +8.7 $mAP_{0.75}$ improvements (from 28.5 to 37.2). This suggests that pseudo-labels generated by EM-PASTE give a strong learning signal for the model to develop object awareness. Lastly, although bounding box is a more insightful cure for instance segmentation, we find our results comparable with the previous SOTAs that use bounding box. Indeed, we are only 0.5 $mAP_{0.50}$ lower compared to the best bounding box SOTA, which requires hand-drawn ground-truth boxes.

Next we demonstrate effectiveness of the proposed EM-PASTE method on MS COCO dataset (Lin et al., 2014b). It is much more challenging than the Pascal VOC dataset as it consists of 80 object classes and each image may contain multiple instances of different classes. Quantitative results are shown in Table 2. We observe that the proposed method can achieve an improvement of +2.8 $mAP_{0.50}$, and +7.5 $mAP_{0.75}$ improvements over previous image-level SOTA. Interestingly, our method with smaller architecture (R-50) outperform prior method IISI (Fan et al., 2018) that works with larger network (R-101). These results provide evidence that our method can scale to large data with large number of object classes.

**Ablation Study** We present the performance of EM-PASTE on PASCAL VOC 2012 val set with different choice of parameters in Table 3. All experiments use R-101 as backbone, training with synthetic data generated by Section 3 and following the details in Section 4.1. We first note that DALL·E is indispensable for best performance, and training only with 20,226 backgrounds from original background pool gives 1.9 lower $mAP_{0.50}$, validating our hypothesis that additional contextual background makes model learn more thorough object representation. Further, it is crucial to choose appropriate backgrounds. A purely black background or a random background[4] does not bring benefit but in turn harm model learning (0.8 and 0.2 $mAP_{0.50}$ lower than not using additional augmented images). Additionally, we quantify the effect of Algorithm 1 by training a model on foreground extracted without F-EM, and observe that iterative foreground refinement is essential for a quality foreground, as F-EM provides 5.0 $mAP_{0.50}$ improvement. Moreover, for the original PASCAL VOC dataset, balanced selection might not work well overall, giving 1.6 $mAP_{0.50}$ lower. Given 30k training set, a balanced selection makes each class approximately 1.5k, and classes with a smaller set of extracted foregrounds will be reused more often, and the potential noise from extraction in Section 3.1 might be amplified. This result suggests a more sophisticated selection method is needed, which we leave for future work. Lastly, the number of paste objects $n_p$ is important in that a value too low results in sparse foregrounds, while a value too large results in crowded foregrounds, each of those hurts the model learning. We empirically show that for PASCAL VOC dataset, 4 seems to be a more appropriate value to use. Surprisingly a fixed 4 gives slightly higher $mAP_{0.50}$ compared to assigning random $n_p \sim \text{Unif}[1, 4]$ dynamically. More analysis on region proposal methods and fore-ground mask quality is in Suppl.

---

[4]For simplicity we use MS COCO images that does not contain any of 20 VOC objects as random background.

Table 3: Ablation study on PASCAL VOC.

| DALL·E | # Paste Objects | Foreground | mAP$_{0.50}$ | mAP$_{0.75}$ |
|---|---|---|---|---|
| ✗ | 4 | - | 56.5 | 35.8 |
| Black | 4 | - | 55.7 | 34.3 |
| Random | 4 | - | 56.3 | 36.7 |
| ✓ | 4 | w/o Algorithm 1 | 53.4 | 35.7 |
| ✓ | 2 | Balanced Selection | 56.8 | 36.3 |
| ✓ | 2 | - | 56.9 | 36.6 |
| ✓ | 1 ∼ 4 | - | 58.0 | **38.1** |
| ✓ | 6 | - | 57.2 | 37.5 |
| ✓ | 4 | - | **58.4** | 37.2 |

## 4.3 Weakly-supervised Object Detection

We argue that EM-PASTE is effective not only in instance segmentation task, but on other tasks as well. We reuse synthesized dataset described in Section 4.1 to conduct object detection on Pascal VOC. We compare our method against two popular baselines, CASD Huang et al. (2020) and Wetectron Ren et al. (2020). In Table 4, we observe that our method achieves almost +4.0 and +5.0 mAP$_{0.50}$ compared to CASD and Wetectron respectively.

Table 4: Object detection on Pascal VOC 2012.

| Method | Backbone | mAP$_{0.50}$ | mAP$_{0.75}$ |
|---|---|---|---|
| Wetectron (Ren et al., 2020) | VGG16 | 52.1 | - |
| CASD (Huang et al., 2020) | VGG16 | 53.6 | - |
| EM-PASTE (Ours) | R-50 | 57.2 | 30.7 |

(a) Long-tail distribution of generated data (Wu et al., 2020). The number of instances for each class shown on the top.

(b) We report mAP@50 for each class. **Gray** values are from Mask RCNN trained directly on data with extracted mask (Section 3.1); values in **red** are the value after EM-PASTE. The classes are ordered the same as (a).

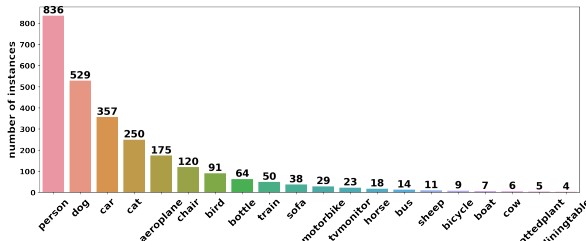

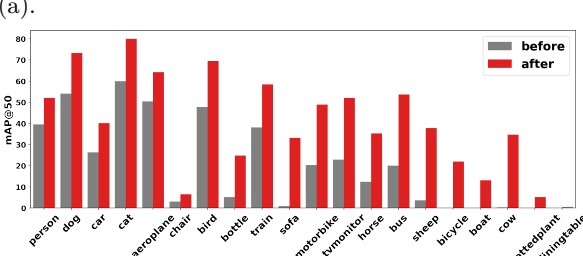

Figure 4: Long-tail instance segmentation setting and results.

## 4.4 Weakly-supervised Instance Segmentation on Long-tail Dataset

We now discuss how EM-PASTE can alleviate the long tail problem. In long-tail (He & Garcia, 2009) dataset, the head class objects contain much more instances than tail classes, so that simple learning method might learn the bias of the dataset, that is, become the majority voter which predicts the head class all the time (Liu et al., 2019; Wang et al., 2021). Due to the ability to generate synthetic data based on selection distribution, even given a highly imbalanced dataset, we can create a synthetic dataset with a balanced class distribution.

**Implementation Detail** We conduct our experiments on a long-tailed version of PASCAL VOC (Everingham et al., 2010) dataset. Our long-tailed dataset, generated based on method proposed in Wu et al. (2020), forces the distribution of each class to follow Pareto distribution (Davis & Feldstein, 1979). It contains 2,415

images in total, with a maximum of 836 and a minimum of 4 masks in a class. Statistics of our generated dataset shown in Figure 4a. The `person` class contains the most instances, while there are 5 classes with less than 10 instances. To the best of our knowledge, we are the first to conduct weakly-supervised instance segmentation task using Wu et al. (2020).

**Results**  We now show that our weakly-supervised instance segmentation can largely mitigate the long tail problem. Our results on PASCAL VOC val set are shown in Figure 4b, with $mAP_{0.50}$ values for each class. We compare with Mask RCNN (He et al., 2017) with details described in Section 4.1, using the long tail dataset itself, i.e. only train with pseudo-labels inferred by Section 3.1. As shown in Figure 4a, training on such an imbalanced data deteriorates the model. Out of 20 classes, there are 10 classes that have $mAP_{0.50} \leq 12.42$, 6 classes that have $mAP_{0.50} < 1$ and 4 classes that are not being recognized by model at all. The overall $mAP_{0.50}$ is 20.26. However, after EM-PASTE with balanced setting (the $p$ in Section 3.3), the model increasing overall $mAP_{0.50}$ to 40.28. All classes show an improvement compared to vanilla training, with an average improvement of 20.0 $mAP_{0.50}$.

## 5  Conclusion

We propose EM-PASTE: an Expectation Maximization guided Cut-Paste compositional dataset augmentation approach for weakly supervised instance segmentation method using only image-level supervision. The core of our approach involves proposing an EM-like iterative method for foreground object mask generation and then compositing them on context-aware background images. We demonstrate the effectiveness of our approach on the Pascal VOC 2012 instance segmentation task by using only image-level labels. Our method significantly outperforms the best baselines. Further, the method also achieves state-of-the-art accuracy on the long-tail weakly-supervised instance segmentation problem.

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
