# OpenReview forum: "EM-Paste: EM-guided Cut-Paste for Image-level Weakly Supervised Instance Segmentation"
_TMLR — Rejected by TMLR_

### Review · Reviewer_aApd · 2023-06-28

**Summary Of Contributions:**

The proposed method, EM-Paste, introduces an Expectation Maximization (EM) guided Cut-Paste compositional dataset augmentation approach for weakly-supervised instance segmentation using only image-level supervision. The method comprises three main components. Firstly, an EM-like approach is presented to refine object mask proposals generated by a region proposal method, resulting in high-quality foreground object masks. These masks are crucial for accurate instance segmentation. Secondly, high-quality context-aware background images are generated using a text-to-image compositional synthesis method, similar to DALL·E. This step ensures that the background context aligns well with the foreground objects, enhancing the overall quality of the dataset. Finally, a large-scale pseudo-labeled instance segmentation training dataset is created by compositing the refined foreground object masks onto the original and generated background images. This dataset enables training a weakly-supervised instance segmentation model.


**Audience:**

Yes

**Claims And Evidence:**

No

**Requested Changes:**


Explanation of EM effectiveness:
Introduction: Provide a detailed explanation of the effectiveness of the Expectation Maximization (EM) algorithm, including theoretical and experimental evidence. Highlight the potential advantages of the EM algorithm in data generation and segmentation tasks.
Methodology: In the method description, provide a clear and comprehensive explanation of how the EM-like approach refines the initial object mask proposals. Discuss the underlying principles and reasoning behind the iterative refinement process. Additionally, discuss any specific experiments or analyses conducted to validate the effectiveness of the EM-guided refinement.
Comparison with more state-of-the-art methods, including Ghiasi (2021):

Related work: Expand the related work section to include a comprehensive review of recent state-of-the-art methods for weakly-supervised instance segmentation, including the work by Ghiasi (2021). Highlight the similarities and differences between EM-Paste and these methods, emphasizing the unique contributions and advantages of EM-Paste.

**Strengths And Weaknesses:**

Strengths:

Performance: The proposed EM-Paste method achieves state-of-the-art results in weakly-supervised instance segmentation on the PASCAL VOC 2012 and MS COCO datasets, surpassing the best baseline by significant margins.
Weaknesses:

Lack of novelty: The concept of copy-pasting and using EM algorithm for mask refinement is not novel and has been explored in previous works.
Limited theoretical understanding: The approach of using EM-like algorithm for mask generation lacks a comprehensive theoretical explanation or justification. The paper does not provide a clear understanding of why this method is effective or how it improves the quality of object masks.
Incomplete comparison: the empirical results only compare the proposed method with a few relatively weak methods, while methods such as Ghiasi(2021) is not listed in empirical comparison.

---

### Review · Reviewer_n38f · 2023-07-20

**Summary Of Contributions:**

The paper proposes an approach to generate a synthetic dataset for training instance segmentation models.

The proposed dataset generation procedure consists of 3 steps:
1. Splitting image into coarse regions and then using a spatial semantic signal from the Grad-CAM model to produce object proposal masks with semantic labels.
2. Generating plausible backgrounds with ruDALL-E model.
3. Pasting object proposals onto the background.

Then the resulting dataset is used for training a Mask R-CNN model.

**Audience:**

Yes

**Claims And Evidence:**

No

**Requested Changes:**

How to make the paper stronger:
  - Address weakness (1).
  - Commit to releasing the synthetic data (or the code to generate synthetic data).
  - Proofread the paper: fix misformatted tables and references, fix other typos.

**Strengths And Weaknesses:**

Strengths:
1. The area of learning dense prediction models from limited supervision is an important research direction.
2. It is a promising idea to use synthetic data that is partially generated with the aid of powerful generative models.

Weaknesses:
1. The main selling point of the paper is that the resulting model does not use segmentation masks. However, one of the models used for obtaining image segments (namely "Convolutional Oriented Boundaries" by Maninis et al) appears to use supervised segmentation data from the PASCAL VOC model. If it is the case, this will invalidate the main experimental results.
2. As the model is quite complex, reproducibility will be a problem. I believe that releasing the resulting synthetic datasets (or code to generate the datasets) may be a valuable contribution that will also unlock the paper's results reproducibility (at least partially).
3. Final results are generally underwhelming. In absolute terms the results are not even remotely competitive with the current SOTA models. Also, in the ablation study, usage of the synthetic background gives only 2 AP points boost.
4. In general, I believe that the constraint of "no single densely labelled mask was used" to be the wrong research direction. First, for virtually any task, it is realistic to acquire 10 to 100 (or more) segmentation masks. These masks can be used to estimate hyper-parameters and other design choices in a principled way and likely will boost the results a lot. Second, I do not believe there are reasonable segmentation models that truly rely on image-level labels only. One way or another, ground-truth segmentation masks affect the design choices that go in the weakly-supervised model. Overall, it is not the reason to reject any publication on this topic, however it constitutes an additional weakness.
5. Related to the previous weakness: how hyper-parameters of such a complex 3-step system (actually 4 steps, if we count the final Mask-RCNN training) were tuned?
6. Paper seems rushed, which is dissapointing given that TMLR does not have a single deadline. Numbers in Table 1 are partially misformatted (e.g. things like "blue!1558.9" or "green!1558.4") or mis-generated references like "COBManinis" in section 3.1.

---

### Review · Reviewer_B3e8 · 2023-08-08

**Summary Of Contributions:**

Instance segmentation, the task of detecting objects and inferring the
pixel-wise masks of their spatial extents, is a visual recognition task that
suffers from prohibitive annotation costs due to the number of objects and
pixels involved. Two existing threads help mitigate the annotation cost of
instance segmentation and the related but simpler task of object detection by 1.
weakening the required supervision by learning from image tags (without
object/instance annotations) or bounding boxes (without pixel-wise annotations)
and 2. synthesizing augmentations of existing annotations by for instance
"cutting and pasting" annotations into different images to make more
combinations of foreground objects and background contexts for training. This
work ties together the two threads by (1) improving the estimation of foreground
object masks from weak image tag supervision by iterative optimization and (2)
synthesizing background images by text-driven diffusion for image synthesis.  In
essence, this work proposes a specialization of DALL·E for Detection (Ge et al.
2021) to instance segmentation.  The method, EM Paste, identifies foreground
object masks from images tagged with ground truth classes, generates background
images without the classes, and then composites the identified foreground masks
into the generated background images for training an instance segmentation
model. The results on MS COCO and PASCAL VOC highlight the accuracy achievable
with only image-level annotations, while also exploring how this measures up to
box-level annotations and how this applies to imbalanced classes. THe method
follows Ge et al. 2021 in the composition of foregrounds and backgrounds that
are separately selected/generated and then combined to augment the training
data.

**Audience:**

Yes

**Broader Impact Concerns:**

None. This work studies an established visual recognition task on standard datasets to reduce the costs of annotation and improve accuracy. This goal is neutral w.r.t. positive or negative impacts that may stem from its use.

**Claims And Evidence:**

Yes

**Requested Changes:**

*Critical Changes for Acceptance*

- Please comment on the cited more recent and stronger box supervision methods
  (BoxTeacher, DiscoBox) and how the claims and evidence of this can be revised
  or not to include them.
- Please clarify Sec. 3.2: in particular, what are the text substitution rules
  for image captioning? Since this presumably makes a difference in the results
  it is worth describing how it is done.
- Please either experiment with space maximization paste or drop its
  description. As it is, a method is partly described but not at all used, which
  is not useful or informative empirically.
- Please scope the experimental claims to the datasets evaluated. The conclusion
  is too broad, and claims the state-of-the-art for long-tail weakly-supervised
  instance segmentation without reference to related work or mention of the
  single dataset reported in the experiments (a variation of PASCAL VOC).
  Sidenote: the conclusion also fails to mention MS COCO although it is reported
  on in the paper, and so deserves inclusion.
- There is a general need for proofreading and editing to correct minor errors.
  See the "Miscellaneous Feedback" section below for suggestions.

*Minor Changes for Improvement*

- Consider grouping the methods in Table 1 not only by backbone but by region
  proposals. This could be too diverse to summarize concisely, but if there are
  some clusters of the same or similar choices then a column could be added to
  Table 1. This would help gauge the effect of region candidates vs.  the other
  components of the proposed method. Alternatively, consider ablating the choice
  of regions for EM-Paste in a separate analysis.
- Discuss unsupervised instance segmentation, such as CutLER, as an even more
  weakly-supervised regime. This work makes the point that image tags are less
  supervised than boxes, so noting that even less superivision is possible (at
  the cost of reduced accuracy) is fair and informative to the reader.
- The introduction should cite R-CNN and SDS for instance segmentation, as these
  precede the given citations, and helped define the task and its evaluation.
- The description of instance segmentation in the introduction could be confused
  for semantic segmentation: is the task per-pixel labeling or object detection
  then segmentation? Importanty, instance segmentation is not just
  pixel-labeling, but the recognition of separate object instances.

*Miscellaneous Feedback*

- Rephrase: change "Entitled Cut-Paste" (Sec. 2) to "This Cut-Paste" because
  "entitled" is not the same as "titled".
- Typo: "in [a] deep learning framework"
- Typo: change "restraint" to "restrain" in "restraint the bounding box by
  tightness"
- Typo: "foregreound" in Figure 2
- Grammar: Please revise the sentence beginning "While," at the top of page 5, under Algorithm 1, to better coordinate its clauses.
- Terminology: Please standardize the choice of "center of " moments / mass / gravity
- Terminology: Please consider alternatives for "location assumption" like "location score" and alternatives to "semantic metric" like "class score" to give these two parallel components more similar names and to reflect their graded levels (because "assumption" sounds absolute).
- Terminology: What does "each with their edges" mean? (Sec. 3.3)
- Terminology: Please rephrase "although bounding box is a more insightful cure" for clarity. Is it that bounding boxes are more precise supervision?
- Formatting: The colors in Table 1 are not formatted correctly and render as the literal text "blue" and "green".



**Strengths And Weaknesses:**

*Strengths*

- This work clearly situates itself in the progression of papers on augmentation
  and weak supervision for instance segmentation, credits main threads like cut
  and paste augmentation, and states specific claims for its benchmark
  improvements.
- There is improvement over methods with the same type of supervision,
  image-level tags, on both MS COCO and PASCAL VOC. The boost to mAP is
  noticeable at two standards of IoU thresholds (0.5 and 0.75) in the 2-5 points
  or more.
- The experiments cover typical datasets and models for comparison with prior
  work. The results are shown on the standard datasets of MS COCO, which is a
  common, and PASCAL VOC, which is the prior gold standard benchmark.  These
  results include boht ResNet-50 and ResNet-101, but no attention architectures,
  although this is fair since the baselines also focus on convolutional
  networks.
- The success of the method provides another point of evidence for the
  usefulness of synthetic data in the training of recognition models. Different
  from Dall-E for Detection, this work shows the same can hold for the more
  spatially-precise task of instance segmentation.

*Weaknesses*

- Experiments: The experimental comparisons do not control for the choice of
  region proposals. These differences could explain away some of the accuracy
  improvements for EM Paste. While it could be a lot of work to re-evaluate each
  baseline, it could be feasible to study this question for one or two, such as
  the strongest prior baselines with image supervision, which is the same
  setting as this work.
- Experiments: The most recent baseline on VOC is from 2021. Furthermore, the
  methods supervised by bounding boxes are older than the methods supervised by
  image tags, which undermines the claim that EM Paste outperforms learning from
  bounding boxes. For instance, BoxTeacher (Cheng et al. CVPR'23) and DiscoBox
  (Lan et al. ICCV'21) both report higher APs on PASCAL VOC along with
  box-supervised results for MS COCO, but neither is cited or compared to here.
  In particular, see Table 2 of each paper for results with ResNet-50, for
  compariability to EM Paste.
- Method: The method and hyperparameter for handling multiple foreground
  segments is heuristic and potentially dataset-specific: a top number of
  segments with minimal distance to the estimated class center are kept. In a
  dataset where the number of instances per image is higher than the chosen
  value, in this case three, it could severely degrade. Experiments or an
  ablation showing an alternative selection criterion that can select more
  segments, such as keeping segments with a given proportion of pixels, would
  help understand how flexible the method is. Similarly, it would be informative
  to report the accuracy binned by the number of instances per image in the
  evaluation sets. Having pointed this out, it could be that the proposed method
  is appropriately balancing precision and recall, since for augmentation
  precision may be more important (so that only correct classes are pasted into
  new images). However, there is no experiment to confirm this.
- Method: Two methods for data augmentation by pasting are proposed, Random
  Paste and Space Maximize Paste, but only Random Paste is included in the
  experiments.  Without experiment, it is not possible to gauge the value of the
  Space Maximize Paste variant, and so it cannot be considered as a contribution
  with a claim and evidence. (See Sec. 3.3. for the description of both.)
- Lack of clarity: In Sec. 3.2, the augmentation of background images by image
  synthesis is not sufficiently explained nor is it experimentally analyzed. In
  particular, what is the rule for subsituting object text to prevent the
  generation of foreground classes. Why assume that synthesized images do not
  contain the foreground, when automatic inspection by a detection model or
  human inspection of even a subset of images could measure if and when
  foreground content is included?  Finally, why not ablate the choice of
  background images by experimenting with pasting onto original images from the
  datasets? This has already been a successful technique, as shown by Dwibedi et
  al. 2017, so it would be valuable to know what image synthesis additionally
  contributes.
- Related work: First of all, the existing related work does a good job of
  situating the submission in the context of data augmentation and weak
  supervision for recognition. However, there is a missing aspect which would
  add valuable perspective alongside box and image supervision: unsupervised
  segmentation learning. In particular, consider citing FreeSOLO (Wang et al.
  CVPR'22. "FreeSOLO: Learning To Segment Objects Without Annotations") and
  CutLER (Wang et al. CVPR'23. "Cut and Learn for Unsupervised Object Detection
  and Instance Segmentation").

*Summary*

This work examines instance segmentation from image-level supervision
and shows a promising degree of improvement over image-level baselines. However,
this contribution is weakened by lack of clarity in some parts of the method,
and its claims are not entirely connected with the evidence due to missing
comparisons with stronger box-supervised methods. Furthermore, experiments could
be more thorough by controlling for more factors like the choice of region
proposals and analyzing assumptions such as the assertion that synthesized
images will not contain foreground classes. Although there are minor
presentation issues (typos, formatting errors, and such) these can be simply
addressed by proofreading. As submitted, the weaknesses outweight the strengths,
but clarifications and rescoping of claims to fit the evidence could make this
appropriately informative to TMLR readers.

Regarding suitability, there is certainly an audience for vision and learning
work on recognition like this submission, but the case for the claims and
evidence in this work is weaker. Please see the requested changes relating to
baselines and scope, as these are important concerns, even though this review
has preliminary marked the claims and evidence as satisfactory ("Yes").

*Questions*

1. Is it possible to express F-EM as a more standard EM iteration with the E-step
  followed by the M-step? It is standard to first estimate the latent variables
  (segments) given the parameters (class-wise mixture models), then maximize
  w.r.t. the parameters given the latents, and iterate. Is there a reason for
  the current ordering of steps?
2. How are region proposals scored by the classifier (the "semantic metric" of
  Section 3.1)? That is, how are the regions preprocessed into inputs for the
  classifier? There could be an effect of how they are masked out or not, how
  they are resized, and so forth.
3. Are there any systematic differences in the loss on real training images vs.
   the proposed synthesized/augmented images? Is there any analysis of the
   generated background images to see if they contain foreground objects,
   whether by manual inspection or filtering by the outputs of a detection or
   instance segmentation model?
4. Related to (3), how does the image captioning substitution or filtering step
   to avoid the generation of foreground objects? Is it simply removing the
   class names from text? This could fail in various ways, such as making
   ungrammatical (and therefore out-of-distribution) captions, or failing to
   remove synonymous object names that still intersect with foreground classes
   (or at least visually resemble the dataset classes).

---

### Decision · Action_Editors · 2023-10-06

**Recommendation:** Reject

**Comment:**

In addition to the lack of evidence mentioned above, the reviewers express several concerns regarding:
- the clarity of some aspects of the method;
- the limited explanation/justification of the method's behavior;
- missing discussion of several related works;
- the complexity of the method, making it potentially difficult to reproduce.
The authors did not respond to the reviewers' comments in the discussion phase, therefore failing to address their concerns.

**Audience:**

Yes.

**Claims And Evidence:**

The reviewers are all unconvinced that the claims made in the submission are supported by clear evidence. In particular, the reviewers would like to see:
- additional experiments to evaluate the effectiveness of some components of the proposed method;
- comparisons to more recent methods;
Furthermore, Reviewer n38f mentions the use of a model that relies on supervised data, which may invalidate the claim of weak supervision.